# Dual-Action Effect of Gallium and Silver Providing Osseointegration and Antibacterial Properties to Calcium Titanate Coatings on Porous Titanium Implants

**DOI:** 10.3390/ijms24108762

**Published:** 2023-05-15

**Authors:** Alejandra Rodríguez-Contreras, Diego Torres, David Piñera-Avellaneda, Lluís Pérez-Palou, Mònica Ortiz-Hernández, María Pau Ginebra, José Antonio Calero, José María Manero, Elisa Rupérez

**Affiliations:** 1Biomaterials, Biomechanics and Tissue Engineering Group (BBT), Department of Materials Science and Engineering, Universitat Politècnica de Catalunya (UPC), Escola d’Enginyeria de Barcelona Est (EEBE), Eduard Maristany 16, 08019 Barcelona, Spain; 2AMESPMTECH, Carrer de Laureà i Miró, 388, 08980 Sant Feliu de Llobregat, Spain; 3Institute for Bioengineering of Catalonia (IBEC), Baldiri Reixac 10-12, 08028 Barcelona, Spain

**Keywords:** porous structures, gallium, silver, biomaterials, antibacterial activity, titanium implants, 3D-printing, coatings

## Abstract

Previously, functional coatings on 3D-printed titanium implants were developed to improve their biointegration by separately incorporating Ga and Ag on the biomaterial surface. Now, a thermochemical treatment modification is proposed to study the effect of their simultaneous incorporation. Different concentrations of AgNO_3_ and Ga(NO_3_)_3_ are evaluated, and the obtained surfaces are completely characterized. Ion release, cytotoxicity, and bioactivity studies complement the characterization. The provided antibacterial effect of the surfaces is analyzed, and cell response is assessed by the study of SaOS-2 cell adhesion, proliferation, and differentiation. The Ti surface doping is confirmed by the formation of Ga-containing Ca titanates and nanoparticles of metallic Ag within the titanate coating. The surfaces generated with all combinations of AgNO_3_ and Ga(NO_3_)_3_ concentrations show bioactivity. The bacterial assay confirms a strong bactericidal impact achieved by the effect of both Ga and Ag present on the surface, especially for *Pseudomonas aeruginosa*, one of the main pathogens involved in orthopedic implant failures. SaOS-2 cells adhere and proliferate on the Ga/Ag-doped Ti surfaces, and the presence of gallium favors cell differentiation. The dual effect of both metallic agents doping the titanium surface provides bioactivity while protecting the biomaterial from the most frequent pathogens in implantology.

## 1. Introduction

The importance of finding new alternatives to protect surgeries from bacteria or other microorganism infections is already well established [1,2,3,4,5], especially nowadays with the appearance of new diseases and microorganisms such as SARS-CoV-2 (Severe Acute Respiratory Syndrome Corona Virus). Around 20% of implant surgeries fail because of microbial infections and 18% due to the mechanical loosening of the implant [1,2].

Due to the increasing antibiotic resistance and the lack of new antibiotics on the market, metal-based material is an important challenge in antibacterial therapy [6,7,8,9]. There are metallic ions that have shown not only appealing antimicrobial effects but also provide certain bioactivity to implant surfaces. For instance, trivalent gallium (Ga) presents some special properties such as an antibacterial effect since it shows similarity with Fe^3+^. Ga^3+^ interferes with the iron metabolism (biological processes requiring ferric cofactors) of a wide range of bacteria. Ga^3+^ cannot participate in the redox reactions and, when substituting for the Fe^3+^ in the enzyme active site, renders it inactive [6,7]. Gallium-based compounds have been used successfully to treat bacterial infections both in vitro and in vivo against several important bacterial pathogens [10,11,12]. Furthermore, it has been proven that Ga is capable of reducing bone resorption by inhibiting osteoclastogenesis, preventing bone calcium release and promoting the osteoinduction of osteoblasts [13,14,15,16], without appearing to be cytotoxic to osteoblasts, nor to inhibit cellular metabolism. Therefore, the applications of Ga to dope biomaterial surfaces are gaining great interest.

Another example of metallic ions used in biomedicine is silver (Ag), which has proven to possess a strong inhibitory and antibacterial effect. In addition, Ag has been applied as an antiviral agent [17]. One of the mechanisms by which silver becomes antibacterial is based on the tendency of the Ag^+^ ion to bind to the thiol groups of bacterial enzymes. This binding interferes with DNA replication [18]. Moreover, the electrostatic attraction between the negatively charged cell membrane of microorganisms and the positively charged Ag^+^ ions can destroy the cell membrane [19]. Although silver is toxic to microorganisms, it is much less dangerous to mammalian cells than other metals [20]. However, bacterial resistance to Ag is growing [21]. Recently, it has been demonstrated that a promising strategy to fight against antimicrobial resistance is combination therapy [22]. The combination of different metallic ions, with different antimicrobial mechanisms, leads to synergistic effects and improves microbial killing, and allows the reduction of potential side effects to the host [23]. 

In previous work, we have developed titanium (Ti) coatings on porous three-dimensional (3D) printed Ti samples doped with Ga [5] or Ag [24], both combined with calcium (Ca). The processes were based on the thermochemical treatment reported by Kokubo and colleagues [21,25,26]. The modifications proposed allowed the incorporation of the ions in the form of Ga-containing calcium titanate and Ag-containing calcium titanate in a more efficient way. While the presence of Ga on the Ti surfaces promoted the antibacterial effect against gram-positive strains, contributed to human osteoblast-like cell (SaOS-2) adhesion, proliferation, and differentiation, and considerably increased the mineralization levels, the incorporation of Ag almost entirely inhibited the bacterial adhesion and proliferation of both the gram-positive and -negative strains tested (*Pseudomonas aeruginosa*, *Escherichia coli*, *Staphylococcus aureus*, and *Staphylococcus epidermidis*). Furthermore, the presence of Ca contributed to the apatite precipitation on the surface of the entire porous structure when incubated in simulated body fluid.

The objective of the present work is to simultaneously introduce Ga and Ag ions in the calcium titanate structure of porous Ti samples by means of some modifications in Kokubo’s thermochemical treatment. The use of calcium acetate instead of calcium chloride, proposed in previous studies [5,24], has made it possible to simultaneously dope Ca and Ag ions into calcium titanate coating unifying its production in a sole step. Moreover, due to Ga having a greater half-maximum cytotoxic concentration relative to Ag [27], it allowed for the reduction of the necessary dose of silver, thereby avoiding possible toxicity.

Due to the rapid increase of nosocomial infections as well as musculoskeletal diseases such as fractures, osteoporosis, and bone metastases with age, effective dual osseointegration and antibacterial implants are urgently required.

## 2. Results and Discussion

In a recently published study [24], Ag was added to the surface of porous Ti, by means of a modification of the thermochemical treatment of Kokubo [25,26] for the formation of calcium titanate, aiming to provide the material with antibacterial properties. In another study [5], gallium was incorporated into the calcium titanate coating to provoke a double effect: an improvement in osteoblastic mineralization and an endowment of some antibacterial effect. Following this double functionality, in this work, Ag and Ga were added together to take advantage of the benefits of both elements. The Kokubo treatments already published in the literature (Figure 1a) and the Kokubo treatments proposed in this work (Figure 1b) are summarized below.

### 2.1. Generation of Bioactive Ga-Containing Calcium Titanate from Calcium Acetate

The objective of this study is the incorporation of gallium and silver simultaneously in calcium titanate coatings in a single process step. This objective is not feasible if calcium chloride is used, as it would precipitate silver chloride [15]. For this reason, the effect of using calcium acetate instead of calcium chloride in the treatment with gallium [5] has been previously evaluated.

The evaluation of the morphology and composition of the CaGa5 sample (Figure 2) showed a porous acicular structure similar to the reported feather-like structure from samples produced by CaCl_2_ (Figure 2a) [26]. The EDS analyses confirmed the presence of both Ga and Ca ions on the surface, and the obtained percentages showed no significant differences compared to the results obtained in these previous studies [5] (Figure 2a,b). The formation of the hydrogenated calcium titanates with Ga was confirmed (Figure 3c) with the characteristic peaks at 276, 430, and 715 cm^−1^, in accordance with the RAMAN spectra resulting from the CaGa5 surface [28]. To verify their bioactivity, the calcium phosphate precipitation test (ISO 23317-2014) [29] was performed. The formation of a dense layer of apatite crystals, forming spherical elements and covering the entire 3D-printed Ti surface (inner and outer) was observed after 7 days in SBF immersion (Figure 3d). The layer was composed of oxygen, phosphorus, and calcium according to the EDS analysis, and the ratio between Ca/P was 1.68 similar to the structure of crystalline hydroxyapatite [5,30].

Thus, these results confirmed that changing calcium chloride for calcium acetate in the treatment with gallium does not affect the formation of the calcium and gallium titanate layer, nor does it affect the bioactivity of the samples.

### 2.2. Reaction of Calcium Acetate with Different Concentrations of Gallium and Silver Nitrate

#### 2.2.1. Surface Characterization

Electron microscopy evaluation of the Ti surfaces treated with the different concentrations of Ag and Ga nitrates showed typical porous acicular feather-like structures (Figure 3a). Thus, the incorporation of metal ions does not affect the typical topography of the treatment. In previous papers, when the surfaces were only doped with silver, a slight increase in the thickness of the feathers of the structure was observed [24]. However, in the case of the simultaneous incorporation of both gallium and silver, this effect is not observed.

Semi-quantitative EDS analyses confirmed the presence of Ca, Ga, and Ag on the surface (inner and outer) of the porous samples (Figure 3b). For the same concentration of silver nitrate, when the gallium nitrate concentration was increased two-fold in the process, the atomic percentage of Ga in the surface doubled, and the Ag atomic percentage was maintained (compare CaGa5Ag2 and CaGa10Ag2 in Figure 3b). The atomic percentage was duplicated on the surface when silver nitrate was increased by only one unit (compare CaGa5Ag1 and CaGa5Ag2 in Figure 3b). 

For the physicochemical characterization of the samples, RAMAN analyses were performed before the heat treatment (Figure 1) since, after this phase, the rutile and anatase peaks become more intense and overlap all other species formed on the surface [5,24]. The spectra obtained were very similar for the different metallic concentrations tested (Figure 3c), showing the main peaks corresponding to both calcium hydrogen titanate (CaHTi) or gallium-hydrogenated calcium titanate (GaCaHTi) at 276, 435, 703, and 875 cm^−1^ [24,31]. The two peaks at 100 cm^−1^ and 150 cm^−1^ were representative of the vibratory lattice mode for Ag2O (95 cm^−1^ and 148 cm^−1^) or for AgCH3COO (silver acetate, 102 cm^−1^ and 140 cm^−1^) [32]. Since C was not detected in the EDS analyses (Figure 4b), the presence of silver acetate could be discarded.

When Ti surfaces were subjected to thermal treatment, hydrogenated titanates oxidized, giving rise to oxides with the presence of Ga in their crystal lattice and water evaporates [5]. Thus, the two peaks at 25 and 48 degrees on the XRD analyses can be attributed to gallium-containing calcium titanate with the following possible stoichiometries: GaxCa1-1.5xTi4O9, GaxCa1-1.5xTi2O4, GaxCa1-1.5xTi4O5 [5,28,33]. Besides, two peaks were observed at 38 and 44 degrees corresponding to metallic silver [24,34]. Previously, nanoparticles of metallic silver were detected within the titanate coating produced by the modification of the thermal treatment adding silver nitrate (AgNO_3_) [24]. It is most likely that these same nanoparticles were also generated in this new modification.

Altogether, this characterization confirmed the presence of gallium and silver in the coating. Gallium is contained in the calcium titanate, while silver is present in its metallic form.

#### 2.2.2. Bioactivity

Porous samples treated with different concentrations of gallium and silver nitrates were incubated for 7 days in SBF, and the surfaces were analyzed under an electron microscope. All samples showed a layer of precipitated calcium phosphate crystals with heterogenous nodule sizes around 2 µm thick. The formation of these crystals and their thickness is in accordance with the results of the previously reported studies (Figure 4a) [5,24]. In previous works, it was observed that the thickness of the apatite layer enlarged when the gallium nitrate concentration increased. However, this difference was evident for high concentrations (100 mM) [5]. This difference in thickness is not observed for concentrations of 5 to 10 mM. 

Thus, the bioactivity effect of the treated surfaces was maintained regardless of the concentration ratio used.

### 2.3. Ion Release and Cytotoxicity

The Ca, Ga, and Ag release curves showed an initial fast release and a stabilization of almost all curves after 24 h of incubation (Figure 4b). In general, the Ag and Ga release from porous 3D Ti structures was favored due to the greater specific surface of the samples compared to the discs, except for the CaGa5Ag1 sample, which is approximately the same release from both 2D and 3D samples. The Ti surfaces produced with a higher concentration of Ga and Ag exhibited greater release of the respective ions, agreeing with the literature where similar studies were carried out under the same conditions (ISO-10993-12 standard, [31]) applying Ga [5] and Ag [24] separately.

Ca curves did not stabilize in the period tested (48 h). This continuous Ca release helped the precipitation of calcium phosphate crystals, reflected in the bioactivity study performed after more than 3 days of immersion in SBF (Figure 5a). 

Regarding the silver release, the porous sample treated with the maximum silver nitrate concentration (CaGa5Ag5) showed greater ion release, and its curve did not stabilize after 48 h. This sample was the only one showing less than 70% cell viability in the toxicity test (Figure 4c). As it was considered cytotoxic, this sample was discarded in the cell study.

Therefore, we can ensure that the samples do not present cytotoxicity at concentrations ≤ 2 mM AgNO_3_ and gallium nitrate ≤ 10 mM Ga(NO_3_)_3_.

### 2.4. Antibacterial Effect

The concept of using different metallic ions with antibacterial properties is not recent. For years, pharmacology has implemented therapies with the synergistic use of antibiotics to increase the efficacy of the treatment [21,22]. The Ag-Ga system can offer good results since Ag ions destabilize the biofilm matrix, increasing the possibility that other ions such as Ga penetrate and act on the bacterial response [35]. 

#### 2.4.1. Bacterial Inhibition Halo

Although the growth inhibition halo assay is approximate and is related to the diffusion of the metallic ion through the agar, it provides information on the antibacterial capacity of surfaces. While the 3D porous samples treated with gallium nitrate (CaGa5) showed antibacterial activity only against the *P. aeruginosa*, the treated samples with both gallium and silver nitrates showed growth-inhibition halos for the four bacterial strains tested (Figure 5a). Relating the Ga ion release results and the minimum inhibitory concentration (MIC) for this strain, the concentration of ions released by the CaGa5 sample is higher than the MIC of *P. aeruginosa* [36]. Thus, the addition of Ag to the Ga-containing samples increased their antibacterial character since the spectrum of bacterial inhibition is greater.

#### 2.4.2. Bacterial Adhesion 

The antibacterial capacity of all the treated samples is confirmed (Figure 5b). Although there were no significant differences in bacterial adhesion between the samples treated with the different concentrations of gallium nitrate and silver nitrate, there was a trend in which bacterial adhesion was lower when the presence of both Ga and Ag ions was greater. The strain *P. aeruginosa* was the most resistant microorganism; however, there was a remarkable decrease in its adhesion when the Ga concentration was 10 mM (CaGa10Ag2), explained by the vulnerability of the strain to Ga [36]. These results suggest that while silver provides a wide spectrum of bacterial inhibition, gallium affords a better response against *P. aeruginosa*, which represents 10% of all microorganisms involved in hip replacement infection [37,38].

In terms of a medical device, the possible toxicity of silver is a concern at the regulatory level. As the inhibitory power of gallium is less than silver, the aim of using both metallic ions is to be able to reduce the concentration of silver needed. Moreover, Ga has other advantages such as stimulating bone formation by acting on osteoblast cells and is already on the US FDA approved list.

### 2.5. Cell Response

Human osteoblast-like cells (SaOS-2) were used to study cell adhesion, proliferation, and differentiation on the Ti surface treated with different Ga(NO)_3_ and AgNO_3_ concentrations. Overall, osteoblastic cell morphology was similar on all surfaces, exhibiting well-developed cytoskeleton and nuclei, and they spread out in a similar way in all cases (Figure 6a). After 24 h of culture (Figure 6b), no significant differences were found in the number of cells adhered on the surface, except for the one with the maximum Ag concentration. The CaGa5Ag2 sample slightly reduced the number of adhered cells in the first 24 h, by 21% in comparison with Ti, this being related to the susceptibility of the cells to the presence of high amounts of Ag.

Regarding osteoblast proliferation, the cell number was measured on days 1, 3, 7, and 14 (Figure 6c). After 14 days of incubation, significant differences between the control and the treated samples were found. All samples showed greater cell numbers compared to the untreated Ti, except for the sample incorporating the maximum Ga concentration, CaGa10Ag2, which cell proliferation was significantly lower. However, a great increase in ALP level was obtained for this sample compared to the untreated Ti control and the other samples treated. This result is in accordance with the differentiation process by which pre-osteoblast cells reduce its proliferation in order to induce gene expression and promote the differentiation to mature osteoblast. 

Therefore, while a lower proliferation value was shown with the highest Ga-doped sample, CaGa10Ag2 (10 mM Ga(NO)_3_), differentiation was strongly promoted, as higher levels of ALP activity were quantified (Figure 6d). 

It is important to take into consideration that the thermochemical treatment created a microtopography (Figure 3a), which can affect cell adhesion and proliferation [5]. Thus, the sample treated with calcium also showed an increase in cell adhesion compared to the untreated control due to this microroughness. This increase is similar to the samples treated with a lower Ga concentration, CaGa5Ag1 and CaGa5Ag2, suggesting that the presence of Ga in the Ti surface does not interfere with cell adhesion and proliferation until it reaches a threshold. This threshold is evidenced in the sample treated with 10 mM Ga(NO)_3_.

## 3. Materials and Methods

### 3.1. Titanium Porous Samples

Porous Ti cylinders 10 mm in diameter and 10 mm high with a macroporosity of 347 ± 1 µm and a microporosity 8.6 ± 0.2 µm were provided by AMES Medical (Catalunya, Spain) and produced via 3D printing using DIW (direct ink writing) technology (Figure 7) [39].

### 3.2. Titanium Porous Samples

Previous to their use, the samples were ultrasonically cleaned with acetone, isopropanol, and ultimately with distilled water for 15 min each. In the first stage, samples were soaked in 5 M NaOH solution (10 mL/sample) at 250 rpm for 24 h at 60 °C. Then, the samples were gently washed with distilled water for 5 min in ultrasounds. In the second stage, samples were immersed in a mixed solution of 10 mL calcium acetate solution (100 mM) and 10 mL nitrate solutions. This nitrate solution was composed of 5 mL of gallium nitrate (X mM (Ga(NO_3_)_3_) and 5 mL of silver nitrate (Y mM AgNO_3_) (Table 1). The concentration used was chosen based on the previous works with Ga [5] and Ag [24].

Afterward, the samples were washed with distilled water for 1 min in ultrasounds and then dried with nitrogen gas. The third stage consisted of thermal treatment of 600 °C for 1 h with an ascending ramp of 5 ºC/min. The samples were cooled down at room temperature. In the fourth stage, Ti samples were immersed in 20 mL/sample in distilled water for 24 h at 80 °C and 250 rpm. Subsequently, they were dried with nitrogen gas and stored in a vacuum.

### 3.3. Bioactivity Characterization

Simulated body fluid (SBF) was prepared according to the ISO 23317:2014 standard [29] by dissolving reagent-grade NaCl, NaHCO3, KCl, K_2_HPO_4._3H_2_O, MgCl_2._6H_2_O, CaCl2, and Na2SO4 in ultra-pure water, and buffered at pH = 7.40 [27,28]. Treated Ti structures were immersed in filtered SBF (25 mL) and incubated at 37 °C for 3, 5, and 7 days. The fluid was renewed every 48 h, and the samples were gently rinsed with distilled water after the treatment. Finally, they were dried in a stove at 40 °C and stored in the desiccator for further evaluation. The coating obtained on the surfaces was evaluated using FESEM, and the layer thickness was determined using high-resolution FESEM imaging and Java-based image processing ImageJ.

### 3.4. Surface Characterization

#### 3.4.1. Field Emission Scanning Electron Microscopy (FESEM)

Ti surface topography generated by modifications of the thermochemical treatment and the formation of apatite on the surface after immersion in SBF was examined with a JSM-7001F scanning electron microscope (JEOL, Toyo, Tokyo, Japan), operating at a voltage of 10 kV. The chemical composition of the surfaces was analyzed using an energy-dispersive X-ray analyzer (EDS, JSM-6400 JEOL) at 20 kV. For the examination of the apatite formation, a Pt-Pd coating was used to provide conductivity.

#### 3.4.2. Fourier Transform Confocal Laser Raman Spectrometry (Raman)

The chemical structure of the obtained surfaces was analyzed with a confocal Raman microscope (inVia™ Qontor^®^, Renishaw Centrus 2957T2, Gloucestershire, UK) using a regular mode laser with a wavelength of 532 nm and a grating of 2400 L/mm. Spectra were acquired with a 50× *g* magnifications objective, 1 s of exposure time, and 40 accumulations.

#### 3.4.3. X-ray Diffraction (XRD)

The phases constituting the treated Ti surfaces were determined with low-angle X-ray spectroscopy (Bruker D8 advance, Bellica, MA, USA) using an angle range from 20 to 60° with a step size of 0.02° and a step time of 1 s with a sample inclination of 1°. The obtained data were analyzed with the EVA software (Bruker, Bellica, MA, USA). 

### 3.5. Ion Release

The Ca, Ga, and Ag ions released from the treated porous (3D) and smooth (2D) surfaces were evaluated according to the ISO-10993-12 standard [31] and using inductively coupled plasma mass spectrometry (ICP-MS, Agilent 5100 SVDV ICP-OES, CA, USA) for the analyses. As previously described [5], each sample was immersed in 3.5 mL of Hanks’ solution (1 mL of Hanks’ solution per gram of porous material) and incubated at 37.5 °C for 1 h, 8 h, 24 h, and 48 h under mild-stirring conditions. One mL of the incubated Hanks’ solutions was taken, filtered, and diluted 1:10 in 2% nitric solution for ICP-MS assessments. The same volume taken was replaced with Hanks’ solution. The tests were performed in triplicate.

### 3.6. Bacterial Characterization

#### 3.6.1. Antibacterial Halo Assay

The antibacterial halo or well diffusion test was carried out according to ISO 14729:2001 [40,41,42], using four bacterial strains, *Staphylococcus epidermidis* (*S. epidermidis*—CECT 231) and *Staphylococcus aureus* (*S. aureus*—CECT 59) as gram-positive strains, and *Pseudomonas aeruginosa* (*P. aeruginosa*—CECT 110) and *Escherichia coli* (*E. coli*—CECT 101) as gram-negative strains. The overnight-incubated bacteria solutions were diluted with sterile brain heart infusion (BHI) solution to reach an absorbance value of 0.2 at 600 nm (bacterial concentration about 108 CFU/mL). Of these solutions, 100 μL was used to inoculate the Trypticase soy agar (TSA) Petri plates. The agar is punched in the middle of the inoculated surface to introduce treated and untreated (control) Ti porous cylinders, and 10 μL of PBS was added on top of the samples to guarantee a certain degree of humidity inside. The plates were then incubated at 37 °C for 24 and 48 h, and the diameter of the growth inhibition halo was measured. The tests were carried out in triplicate.

#### 3.6.2. Bacterial Adhesion Assay

Bacteria were cultured overnight in BHI medium at 37 °C, and cell suspensions with concentrations equivalent to OD600 of 0.2 were prepared. Sterile-treated and untreated (control) Ti solid discs placed in 24-well plates were inoculated with 1 mL of these solutions and incubated for 2 h at 37 °C for cell adhesion. Afterward, the samples were changed into a new 24-well plate, and they were washed once with PBS. In order to detach bacterial cells from the surfaces, ultrasounds were applied for 1 min in a PBS volume. The PBS volume was recovered, and dilutions were made to proceed to the cell count in agar plates. Then, the colony-forming units (CFU) were obtained and compared with those of the control samples (Ti).

### 3.7. Cell Response

#### 3.7.1. Cell Culture 

SaOS-2 osteoblast-like cells (ATCC, Manassas, VA, USA) were cultured in McCoy’s Medium Modified with 1.5 Mm glutamine (GibcoTM, Carlsbad, CA, USA) supplemented with 15% of fetal bovine serum (FBS), and 1% of penicillin/streptomycin (50 U/mL and 50 μg/mL, respectively) (GibcoTM). Cells were used in all experiments at passage 40 and maintained and expanded at 37 °C in a 95% humidified atmosphere containing 5% of CO2.

#### 3.7.2. Cytotoxicity

The methodology used to analyze the cytotoxicity of treated Ti surfaces was an indirect in vitro test carried out with human osteoblast-like SaOS-2 cells (ATCC, Manassas, VA, USA) according to ISO 10993-5 [43]. Although SaOS-2 cells are an osteosarcoma cell line, their suitability as an osteoblast cell model has been widely demonstrated [23,24,25,26]. Each specimen was immersed (1 mL/g) in Dulbecco’s modified Eagle’s medium (DMEM) for 72 h at 37 °C. Afterward, the extraction medium was removed and diluted with DMEM (dilution 1/0; 1/1; 1/10; 1/100; 1/1000) and added to previously seeded SaOS-2 cells. After 24 h of incubation, cells were lysed with M-PER (Pierce, Appleton, WI, USA), and the activity of the lactate dehydrogenase (LDH) enzyme was analyzed using the Cytotoxicity Detection Kit LDH (Cytotoxicity Detection Kit-LDH, Roche Diagnostics, Mannheim, Germany). Absorbances were recorded at 492 nm using a Synergy HTX Multi-Mode Reader (Bio-Tek, Santa Clara, CA, USA). As a reference for 100% maximum survival, cells were placed in TCPS (tissue culture polystyrene).

#### 3.7.3. Cell Adhesion 

In order to carefully study the effect of the incorporated metallic ions in calcium titanate structures, in vitro cell studies were previously performed on polished titanium discs. In the case of 3D structures, small variations in porosity and/or roughness could make it difficult to discern their effect. Treated and untreated Ti solid discs were sterilized by washing them in ethanol (70%, *v*/*v*) for 20 min and then were rinsed thrice with PBS. Afterward, the samples were incubated in a cell culture medium for 24 h at 37%.

For immunofluorescence staining, SaOS-2 cells were seeded on the surface of Ti samples at a density of 1.5 × 10^4^ cells per sample and incubated for 24 h at 37 °C. After incubation, the cell medium was removed, and the samples were rinsed with 500 μL PBS; then, cells were fixed with 500 μL 4% of paraformaldehyde (Sigma, Saint Louis, MO, USA) for 20 min at room temperature. After fixation, samples were washed thrice with PBS-Gly (20 Mm glycine in PBS) for 5 min. Cells were permeabilized with 0.05% Triton X-100 in PBS for 20 min at RT, rinsed thrice again with PBS-Gly, and blocked with 1% BSA in PBS for 30 min. The medium was removed, and cells were incubated with the primary antibody mouse anti-vinculin (V9131, Sigma-Aldrich, St. Louis, MO, USA) (1:400 in 1% BSA in PBS). After incubation, samples were washed thrice with PBS-Gly and incubated with secondary antibody Alexa Fluor 488 goat anti-mouse IgG (R37120, Invitrogen, Waltham, MA, USA) for 1 h in the dark following the manufacturer’s protocol. The samples were washed with PBS-Gly and incubated with Alexa Fluor 546 Phalloidin-Rodhamine (A22283, Invitrogen, Waltham, MA, USA) (1:400 in PBS-0.05% triton) for 1 h in the dark. Samples were washed thrice with PBS-Gly, and DAPI (1:1000 in PBS-Gly) was added for 2 min in the dark. After incubation, the samples were finally rinsed with PBS-Gly and were ready for image analysis. Six randomized images of each sample were captured using a Zeiss laser scanning microscope (Carl Zeiss AG, Oberkochen, Germany). For cells/cm^2^, the number of cells was quantified using ImageJ software (version 1.53a, Wayne Rasbind) (*n* = 3).

#### 3.7.4. Cell Proliferation

SaOS-2 cells were seeded on the surface of Ti samples at a density of 1 × 10^4^ cells per sample and maintained in culture for 24 h, 3 days, 7 days, and 14 days, changing the medium twice or thrice a week. After each incubation period, the cells were rinsed thrice in 500 µL PBS and incubated with 350 µL 10% of PrestoBlueTM (Invitrogen, Waltham, MA, USA) for 1 h at 37 °C. As a negative control, PrestoBlueTM was incubated without cells. Then, 100 μL of the medium was transferred to a black 96-well plate to measure an excitation wavelength at 560 nm and an emission wavelength at 590 nm using Synergy HTX multimode reader (Bio-Tek, Winooski, VT, USA). 

#### 3.7.5. Cell Differentiation

The same cells used for the cell proliferation assay were employed to quantify the activity of alkaline phosphatase (ALP). The cells were rinsed with 500 µL PBS to remove the remainder of PrestoBlueTM and then lysed with 500 µL of mammalian protein extraction reagent (M-PER, Thermo Fisher Scientific, Waltham, MA, USA). Samples were incubated for 30 min at 37 °C using the SensoLyte pNPP Alkaline Phosphatase Assay Kit (AnaSpec Inc., Seraing, Liège, Belgium), and absorbance was registered at 405 nm using a Synergy HTX multimode reader (Bio-Tek, Winooski, VT, USA). Results were extrapolated from a calibration curve using purified ALP from the kit. The resulting ALP quantity was normalized versus the time of incubation and their corresponding cell numbers obtained in the cell proliferation assay.

### 3.8. Statistical Analyses

Biological results were expressed as a mean value of standard deviation (SD) for each sample. The *t*-test was used with a 95% confidence interval to evaluate statistical differences in means between the two groups.

## 4. Conclusions

This study addresses the importance of finding surface treatments on titanium implants to improve their biointegration. The proposed modification of the thermochemical treatment represents a simple, economical, and biocompatible improvement to achieve this objective in both smooth and porous implants. The dual effect of the simultaneous incorporation of Ga and Ag occurs in two contexts. In terms of the antimicrobial capacity of the surface, this is mainly achieved by the presence of silver. However, the addition of Ga increases the inhibitory effect against *P. aeruginosa*, thus extending the antimicrobial effect of the surfaces. Regarding the in vitro study, gallium was the agent that improved the osteoblast-like cell response and provoked their differentiation. Altogether, the sample generated with 2 mM silver nitrate and 10 mM gallium nitrate (CaGa10Ag2) was the one that showed the most effective antibacterial character and cell response.

After a series of studies on the modification of Kokubo’s thermochemical treatment, by adding gallium, silver, and both metallic ions simultaneously, the next step will address the in vivo studies of the samples treated. The main objective of these studies is the implantation process at an industrial level.

## Figures and Tables

**Figure 1 ijms-24-08762-f001:**
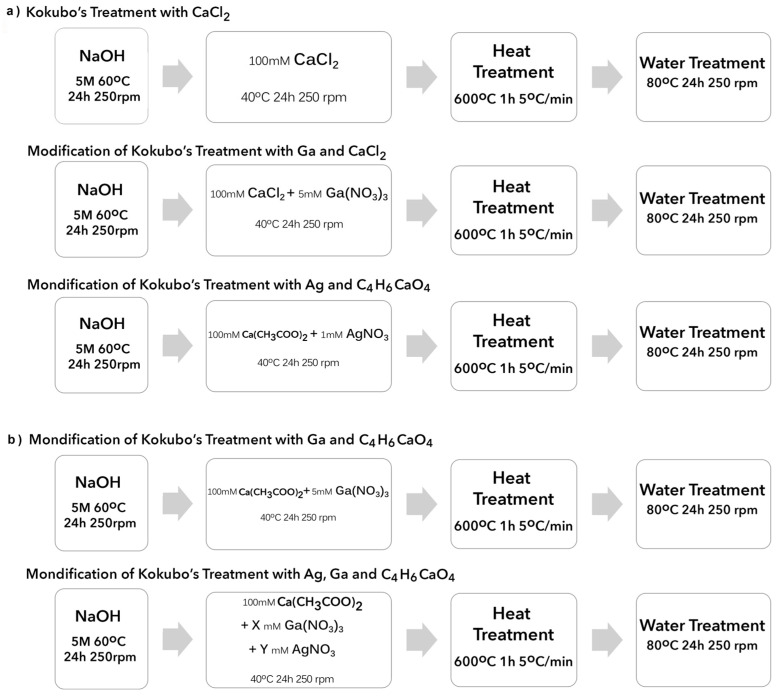
Scheme of the thermochemical treatments on Ti developed in previous studies (Kizuki 2010 [26]), (Rodríguez-Contreras 2020 [5]) (Rodríguez-Contreras 2021 [24]) (**a**) and the ones proposed in this work (**b**).

**Figure 2 ijms-24-08762-f002:**
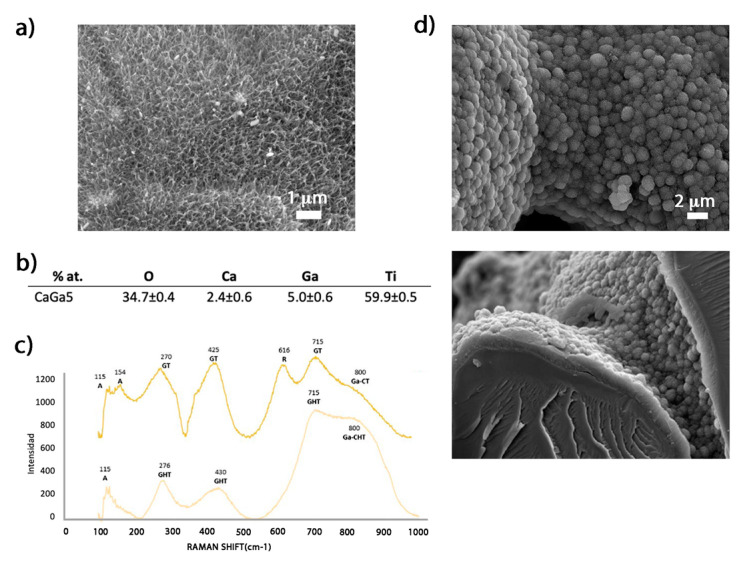
Characterization of the surface treated with calcium acetate and 5 mM gallium nitrate (CaGa5): (**a**) SEM image at 10,000× of the surface. (**b**) Elemental analysis EDS (% at.). (**c**) Raman spectra of the samples treated before (light yellow line) and after (dark yellow line) the thermal treatment. A: anatase, R: rutile, GHT: gallium-hydrogenated titanate, GT: gallium titanate, Ga-CHT: gallium-hydrogenated calcium titanate, Ga-CT: calcium titanate with gallium. (**d**) SEM images at 4000× of the surface (above) and the cross-section (below) of the layer generated after incubating the CaGa5 sample for 7 days in SBF.

**Figure 3 ijms-24-08762-f003:**
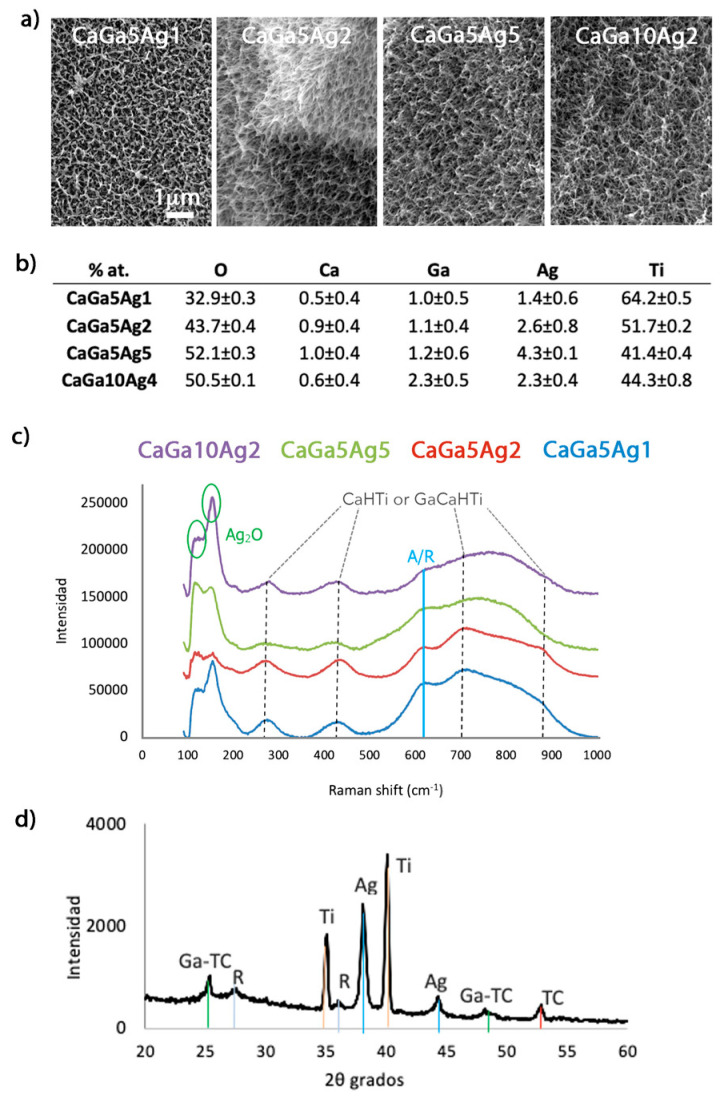
Characterization of the samples treated with calcium acetate and different concentrations of gallium and silver nitrate: (**a**) SEM images of the surfaces at 10,000×. (**b**) Elemental analysis EDS (% at.). (**c**) Raman spectra of the samples (before heat treatment): CaHTi: calcium hydrogen titanate, GaCaHTi: gallium-hydrogenated calcium titanate, A: anatase, R: rutile. (**d**) XRD diagram of a disc sample after completing the thermochemical treatment with calcium acetate and 5 mM of gallium nitrate and 1 mM of silver nitrate with calcium acetate and 5 mM gallium nitrate and 1 mM silver nitrate (CaGa5Ag1). Ga-TC: calcium titanate with gallium and silver, R: rutile, Ti: titanium, Ag: silver, TC: calcium titanate.

**Figure 4 ijms-24-08762-f004:**
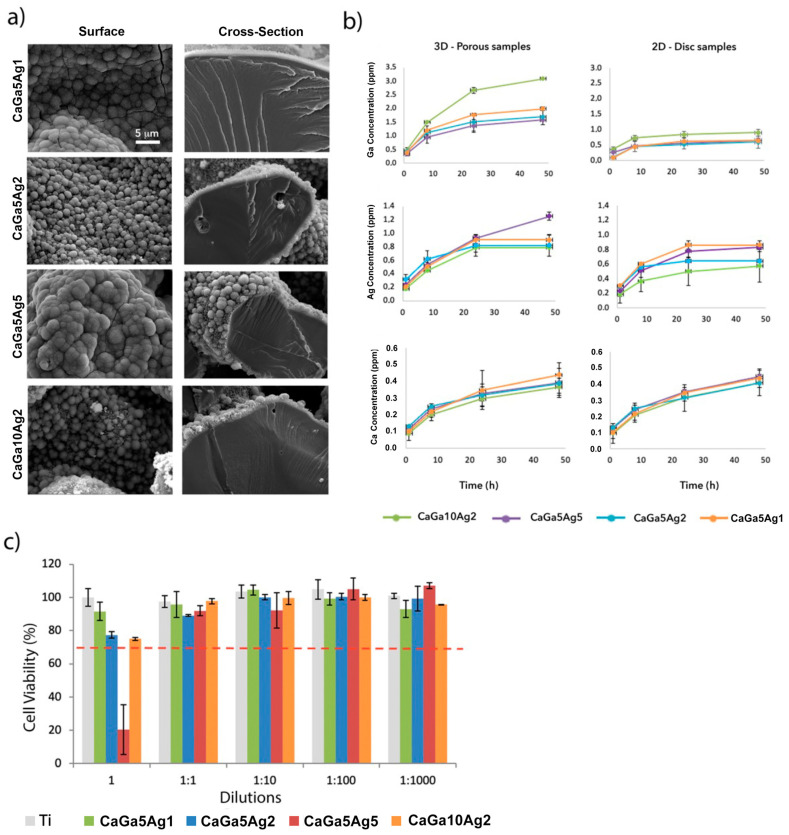
(**a**) SEM images (×4k) of the surfaces of the samples treated with calcium acetate and different concentrations of gallium and silver nitrate incubated in SBF for 7 days. (**b**) Accumulative release of Ga, Ag, and Ca from porous (3D) and smooth (2D) Ti surfaces treated with different concentrations of both gallium and silver nitrate. (**c**) Cell viability of porous samples treated with different concentrations of silver and gallium nitrate, compared with untreated Ti as a control. The red dash line symbolize the limitation of toxicityat 70% cell viability.

**Figure 5 ijms-24-08762-f005:**
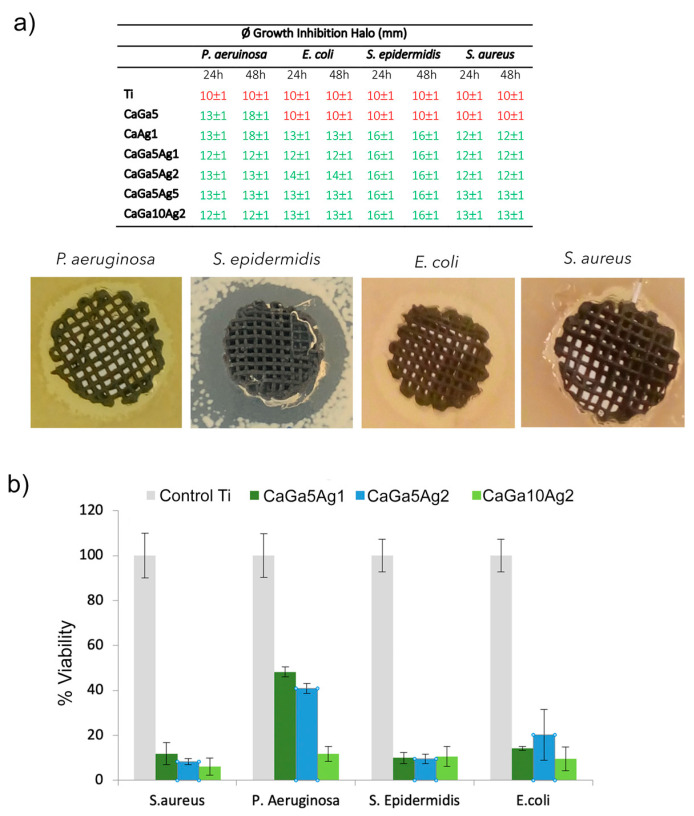
Bacterial response: (**a**) diameter of the growth-inhibition zone generated on Ti sample CaGa10Ag2 after 24 of bacterial incubation (red numbers indicate the value of the control and green number are the values exceeding the diameter of the control); (**b**) bacterial quantification after incubation for 2 h at 37 °C.

**Figure 6 ijms-24-08762-f006:**
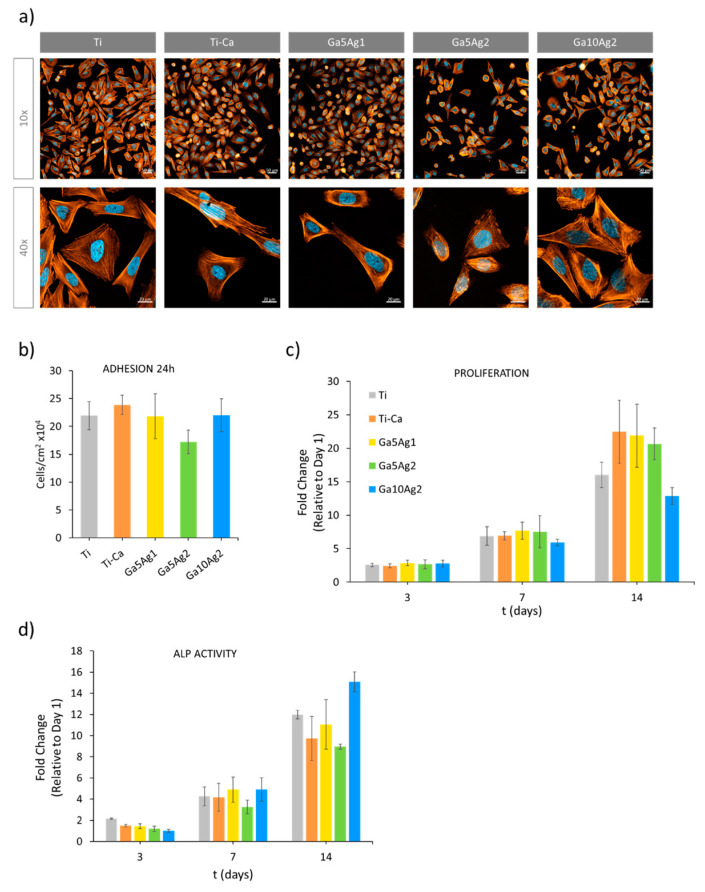
Cell response. (**a**) Representative images of SaOS-2 cells seeded for 24 h using fluorescence staining. Cells were labeled with DAPI (nuclei, blue) and phalloidin (F-actin, orange). (**b**) Quantification of the number of adhered cells per cm^2^ seeded on the surface of titanium samples for 24 h (*n* = 3). (**c**) Proliferation of the cells seeded on the titanium samples for 14 days. Days 1, 3, 7, and 14 were taken as time points to measure the metabolic activity of SaOS-2 (*n* = 3), and the results were expressed as fold change of the control (Ti). (**d**) Quantification of the ALP activity in the cells seeded on the surface of the samples at days 3, 7, and 14 (*n* = 3). Results were expressed as fold change of the control.

**Figure 7 ijms-24-08762-f007:**
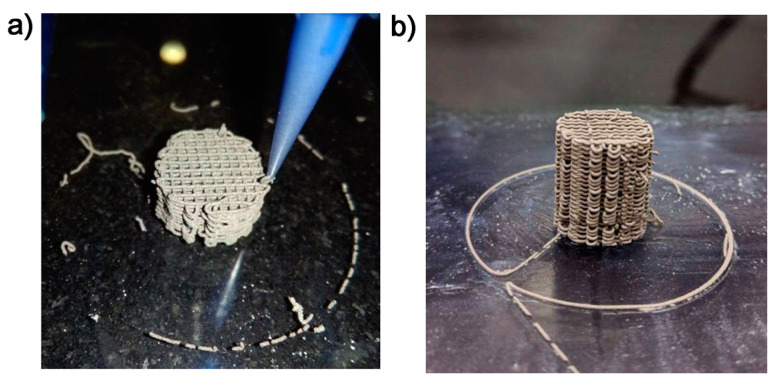
Sample production (images provided by AMES Medical): (**a**) image of the printing process to produce (**b**) porous Ti cylinders.

**Table 1 ijms-24-08762-t001:** Nitrate concentrations for the second stage of the Ti surface thermochemical treatment.

Sample Reference	X Ga(NO_3_)_3_ (mM)	Y AgNO_3_ (mM)
CaGa5	5	0
CaAg1	0	1
CaGa5Ag1	5	1
CaGa5Ag2	5	2
CaGa5Ag5	5	5
CaGa10Ag2	10	2

## Data Availability

Data is unavailable due to privacy or ethical restrictions.

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
