# Peer review of "Dual-Action Effect of Gallium and Silver Providing Osseointegration and Antibacterial Properties to Calcium Titanate Coatings on Porous Titanium Implants"

_ijms, 2023, doi:10.3390/ijms24108762_

Round 1

Reviewer 1 Report

The manuscript under review is aimed at improving the biological characteristics of metal implants. This study builds on previous work of these authors (ref. 22) on the development of modified calcium titanate coatings on 3D porous titanium structures. The main idea is to include gallium and silver ions instead of silver ions only.

Such coatings were prepared from solutions with different concentrations of gallium and silver according to a previously developed method. The samples were characterized by SEM, EDS, Raman-spectroscopy and powder XRD methods. Some in vitro biological characteristics are also presented.

In general, the manuscript contains new scientific results that meet the interests of the readers of IJMS. However, for publication it is necessary that the presentation and discussion of the results be more convincing.

1) In the abstract, the authors state that "The Ti surface doping was confirmed by the formation of Ga-containing Ca titanates and nanoparticles of metallic Ag within the titanate coating". The evidence presented is insufficient.

In particular, an EDS map of the distribution of metals in the coating is needed to assess the content uniformity.

Second, the authors' statement about the formation of metallic silver nanoparticles is incomprehensible. It is not clear which method showed the presence of individual Ag nanoparticles. Their size distribution is unclear. Moreover, although the coatings contain metallic silver according to XRD data, the oxidized silver is found according to Raman spectroscopy.

2) The representation of the composition of the coatings according to the EDS data looks incorrect (Figs. 4b, 3b).

First, it is not clear how the analyzer used can produce such low atomic content errors (e.g, 34.7±0.4 oxygen). This requires a number of special calibrations. No relevant information provided. Without calibration, the errors given by the software are only indicative. In fact, such errors are then not less than 5-10% of the determined value.

Secondly, values of type "0.5±4" (calcium content in Fig. 4b) have no physical meaning. Here, the "determination error" exceeds a certain value. It's just incorrect data.

Moreover, what is the reason for such a low content of calcium, which does not correspond to the stoichiometry of titanate? I mean, why is the calcium content much less than the content of the doping component, for example, silver? Initially, the amount of calcium introduced was at least an order of magnitude greater than that of silver. Explain the reason for this discrepancy also in the text of the manuscript.

3) The absolute absence of carbon in the composition of the coating raises doubts. In fact, such impurities may remain with the chosen method of preparation. It is required to add the EDS spectra in the corresponding area in the Supplementary Materials.

4) The effect of doping components on the morphology of coatings is not discussed. However, the data in Figure 5a suggest that there is some effect. Give particle sizes on the surface and a description of the morphology, note the thickness of the coatings.

5) The data in Figure 5b looks incorrect.

The axes need to be fixed, now they look artificially drawn and do not match the graphs.

It is necessary to explain why "whiskers of error" are present both on the vertical axis (metal content) and on the horizontal axis (time). Should it be understood that in the experiments it was not possible to clearly fix the time?

It is necessary to rid the text of the manuscript of contradictions with the data of the Figure. In fact, judging by the Figure, silver release is not much different for porous and non-porous samples (see scales). It is clear from the text that it is different (limes 345-346).

On the other hand, it follows from the text that saturation (stabilization) is not reached only for calcium release (line 350). However, the purple silver release curve also did not reach saturation. It's not explained.

Comparison of ion release kinetics with literature (lines 347-349) is unclear. Clarification of the experimental conditions is required.

Finally, explain how the gallium content of the solution was measured and how the samples were prepared on the disks. What was the initial morphology of the disks and what was the morphology of the coatings on them. Provide SEM data. Otherwise, the comparison is incorrect.

6) As far as is clear from the experimental part, сell viability was examined after 24-h exposures. Thus, the explanation for the toxic effect observed for sample CaGa5Ag5 is incorrect (lines 354-355). In fact, after 24 hours, the silver concentration there is no higher than in other samples (Fig. 5b). Therefore, offer a more correct explanation.

In addition, specify the limits of the safe concentration of each metal in biological solutions (in ppm, according to the literature data and compare with the content being achieved.

7) According to Figure 6a, the sample CaGa5 as well as CaAg1 has a more pronounced antibacterial effect towards to P. aeruginosa than all the bimetallic-contained samples. Thus, a synergistic effect is manifested in a decrease in bactericidal action. Now this is not consistent with what is written in the text. Please bring the information in line with the data and explain the possible reasons for this effect.

In Figure 6b, there are no data for "monometallic-doped " samples (CaGa5 and CaAg1). In general, from the results of the figure it seems that the "bimetallic-doped" samples (CaGaXAgY) are noticeably less efficient than the monometallic ones.

8) Similarly, section 3.5. Cell Response does not provide comparative data on "monometallic-doped" coatings (CaGa5 and CaAg1). This does not allow us to support the idea and final conclusions of the article about the preference for "bimetallic-doped" coatings.

9) Although the article is titled "Synergistic effect..." no clear evidence of existing synergism was found. If the coating combines the positive properties of the components (Conclusions), then this is not synergism. Synergism necessarily includes the emergence of new properties or a fundamental change in existing ones. This is not observed here, or is not clearly presented.

The reasons for the expected synergies should also be briefly discussed.

10) Technical Note: Calcium acetate notation as "C4H6CaO4" in Fig. 1 is confusing. Please enter the chemical formula, such as for calcium chloride.

Author Response

Reviewer 1:

The manuscript under review is aimed at improving the biological characteristics of metal implants. This study builds on previous work of these authors (ref. 22) on the development of modified calcium titanate coatings on 3D porous titanium structures. The main idea is to include gallium and silver ions instead of silver ions only. 

Such coatings were prepared from solutions with different concentrations of gallium and silver according to a previously developed method. The samples were characterized by SEM, EDS, Raman-spectroscopy and powder XRD methods. Some in vitro biological characteristics are also presented.

In general, the manuscript contains new scientific results that meet the interests of the readers of IJMS. However, for publication it is necessary that the presentation and discussion of the results be more convincing.

The authors want to acknowledge the reviewer for taking the time for the review of our manuscript. We proceed to discuss each point:

1) In the abstract, the authors state that "The Ti surface doping was confirmed by the formation of Ga-containing Ca titanates and nanoparticles of metallic Ag within the titanate coating". The evidence presented is insufficient.  In particular, an EDS map of the distribution of metals in the coating is needed to assess the content uniformity.

We already did an EDS mapping at the time, but no particular agglomeration of any of the metal ions was observed. Thus, we understand that the sensitivity of EDS mapping is not enough to study the distribution of these ions. In a previous work [1], the presence of Ag nanoparticles was confirmed by TEM images of the cross-section of a FIB lamella of the Ti sample treated with only AgNO3. The sensitivity of EDS mapping was not enough.

[1] Bioactivity and antibacterial properties of calcium- and silver-doped coatings on 3D printed titanium scaffolds;https://doi.org/10.1016/j.surfcoat.2021.127476).

Second, the authors' statement about the formation of metallic silver nanoparticles is incomprehensible. It is not clear which method showed the presence of individual Ag nanoparticles. Their size distribution is unclear. Moreover, although the coatings contain metallic silver according to XRD data, the oxidized silver is found according to Raman spectroscopy.

Ag nanoparticles were observed and analyzed in a previous work of ours [1]. The XRD results show the same metallic silver presence observed before. We clarified this in the third paragraph in section 3.2.1. Surface characterization

Basically, according to Ellingham diagrams, the silver oxide present before heat treatment (Raman studies) decomposes at temperatures above 200 ◦C. Therefore, after the thermal treatment of the process when 600 ◦C was applied, the metallic silver appeared as a product of the decomposition of Ag2O. Consequently, no Ag oxides were observed in the XRD spectrum after thermal treatment.

2) The representation of the composition of the coatings according to the EDS data looks incorrect (Figs. 4b, 3b).  First, it is not clear how the analyzer used can produce such low atomic content errors (e.g, 34.7±0.4 oxygen). This requires a number of special calibrations. No relevant information provided. Without calibration, the errors given by the software are only indicative. In fact, such errors are then not less than 5-10% of the determined value.

Secondly, values of type "0.5±4" (calcium content in Fig. 4b) have no physical meaning. Here, the "determination error" exceeds a certain value. It's just incorrect data. 

We agree with the reviewer about the values of the standard deviations, and we corrected the errors. The objective of EDS here is only to identify the incorporation of Ga and Ag ions in the titanate surface and to have an idea of the quantity. We are aware of the error of the technic and that’s why the results are semiquantitative. However, we could confirm that, for example, when an element was doubled in the initial concentration applied, the detection increases with the EDS.

Moreover, what is the reason for such a low content of calcium, which does not correspond to the stoichiometry of titanate? I mean, why is the calcium content much less than the content of the doping component, for example, silver? Initially, the amount of calcium introduced was at least an order of magnitude greater than that of silver. Explain the reason for this discrepancy also in the text of the manuscript.

It is not a stoichiometric structure, since the ions are incorporated into the interstices of the titanates [2]. It is a deficient structure of Ca, Ag and Ga, due to the last process of heat water, which improves their release.

[2] Takashi Kizuki et al. Preparation of bioactive Ti metal surface enriched with calcium ions by chemical treatment doi:10.1016/j.actbio.2010.01.007

3) The absolute absence of carbon in the composition of the coating raises doubts. In fact, such impurities may remain with the chosen method of preparation. It is required to add the EDS spectra in the corresponding area in the Supplementary Materials.

We eliminate the carbon, to have the absolute values of the ions to study. The values we obtained for C were between 1 and 0.5 % but considering the errors it is not significant. EDS helps us to have an approximation.

4) The effect of doping components on the morphology of coatings is not discussed. However, the data in Figure 5a suggest that there is some effect. Give particle sizes on the surface and a description of the morphology, note the thickness of the coatings.

It was discussed but in section 3.2.1.: “Electron microscopy evaluation of the Ti surfaces treated with the different concentrations of Ag and Ga nitrates showed typical porous acicular feather-like structures (Figure 4a).” And we incorporated the following sentence: “Thus, the incorporation of metal ions does not affect the typical topography of the treatment “.

We have to clarify that the data in Figure 5a correspond to the images of the SBF essay, showing a Hydroxyhypatite coating, which is commented on in section 3.3. Ion release and cytotoxicity.

5) The data in Figure 5b looks incorrect. The axes need to be fixed, now they look artificially drawn and do not match the graphs.

The authors considered the increase of the axes’ letter size to adapt better the edition of the paper and facilitate its reading. Logically, if necessary, it can be changed.

It is necessary to explain why "whiskers of error" are present both on the vertical axis (metal content) and on the horizontal axis (time). Should it be understood that in the experiments it was not possible to clearly fix the time?

In principle, time has to be fixed. However, in some cases due to experimental errors, there were time variations that we wanted to reflect in the graph.

It is necessary to rid the text of the manuscript of contradictions with the data of the Figure. In fact, judging by the Figure, silver release is not much different for porous and non-porous samples (see scales). It is clear from the text that it is different (limes 345-346). 

We introduce this observation in the text. “In general, the Ag and Ga release from porous 3D Ti structures was favored due to the greater specific surface of the samples compared to the discs, with the exception of the CaGa5Ag1 sample, which is approximately the same release from both 2D and 3D samples.”

On the other hand, it follows from the text that saturation (stabilization) is not reached only for calcium release (line 350). However, the purple silver release curve also did not reach saturation. It's not explained.

We introduce the observation in the test: “The Ag release curve with the maximum Ag concentration (CaGa5Ag5) did not reach saturation in the time of the assay. This can be an advantage in its application as an antibacterial agent”.

Comparison of ion release kinetics with literature (lines 347-349) is unclear. Clarification of the experimental conditions is required.

The cited articles are works of ours and were tested under the same conditions according to the ISO-10993-12 standard. Thus, they are comparable.

We clarify this in the test: “The Ti surfaces produced with the higher concentration of Ga and Ag ions exhibited greater release of the respective ions agreeing with the literature where similar studies were carried out, applying Ga [5] and Ag [22] separately.“

Finally, explain how the gallium content of the solution was measured and how the samples were prepared on the disks. What was the initial morphology of the disks and what was the morphology of the coatings on them. Provide SEM data. Otherwise, the comparison is incorrect.

This is all in M&M:

  • 5. Ion release: by means ICP analyses, the containing of Ga was analyzed.
  • 2. Surface thermochemical treatment: here, you will find how the samples with Ga were prepared on disks.
  • Figure 3 and 4: the morphology of the surfaces treated with ga is shown.

6) As far as is clear from the experimental part, сell viability was examined after 24-h exposures. Thus, the explanation for the toxic effect observed for sample CaGa5Ag5 is incorrect (lines 354-355). In fact, after 24 hours, the silver concentration there is no higher than in other samples (Fig. 5b). Therefore, offer a more correct explanation.

In the viability test, the samples with the thermochemical treatment were incubated for 72h in Dulbecco's modified Eagle's medium (DMEM). Then, cells were cultured in this medium. This means that the ion released to which cells were exposed was the equivalent of 72h. Probably, the continuous release of silver ions can produce toxicity.

In addition, specify the limits of the safe concentration of each metal in biological solutions (in ppm, according to the literature data and compare with the content being achieved.

There is a consensus on the MIC Ag ions values for different bacterial strains. In our case, we exceed the MIB values reported in the literature. The release curves show that the silver ion values at 24h oscillate between 0.8-1 ppm (mg/L). In all cases, we exceed the MIC values reported in the literature; Staphylococcus aureus MRSA [3] (0.84 mg/L), Pseudomonas aeruginosa [4] (0.84-0.85 mg/L),Escherichia Coli [4] (1 mg/L) and Staphylococcus epidermidis [4] (0.5 mg/ L).

[3] Libor Kvitek et al. Antibacterial activity and toxicity of silver- nanosilver versus ionic silver DOI:10.1088/1742-6596/304/1/012029

[4] Wen-Ru Li et al. A comparative analysis of antibacterial activity, dynamics, and effects of silver ions and silver nanoparticles against four bacterial strains. DOI: 10.1016/j.ibiod.2017.07.015

7) According to Figure 6a, the sample CaGa5 as well as CaAg1 has a more pronounced antibacterial effect towards to P. aeruginosa than all the bimetallic-contained samples. Thus, a synergistic effect is manifested in a decrease in bactericidal action. Now this is not consistent with what is written in the text. Please bring the information in line with the data and explain the possible reasons for this effect. 

The growth inhibition halo assay is approximate and is related to the diffusion of the metallic ion through the agar. Thus, the results from this essay give us some information about the metallic ion antibacterial capability. The small differences in halo diameters are not significant. We clarify this in the test.

In Figure 6b, there are no data for "monometallic-doped " samples (CaGa5 and CaAg1). In general, from the results of the figure it seems that the "bimetallic-doped" samples (CaGaXAgY) are noticeably less efficient than the monometallic ones

Probably when carrying out the study of bacterial adhesion at such a short time of 2h, a clear synergistic effect is not appreciated. The antibacterial effect of gallium ions is less than in the case of silver. For example, the MIC values reported for gallium ions are higher than for silver; for example E. Coli (256 mg/L), Pseudomonas aeruginosa, P. aeruginosa and Staphylococcus aureus aprox. 512 mg/L (5).

In terms of a medical device, the possible toxicity of silver is a concern at the regulatory level. As the inhibitory power of Gallium is less than silver, the aim of using the synergy of both metallic ions is to be able to reduce the concentration of silver needed. Moreover, Ga has other advantages as to stimulate bone formation by acting on osteoblast cells and is already on the US FDA approved list.

[5] Antimicrobial effect of gallium nitrate against bacteria encountered in burn wound infections. DOI: 10.1039/C7RA10265H

8) Similarly, section 3.5. Cell Response does not provide comparative data on "monometallic-doped" coatings (CaGa5 and CaAg1). This does not allow us to support the idea and final conclusions of the article about the preference for "bimetallic-doped" coatings.

Yes, we could indeed have tested the two metals separately. However, this assay was carried out to assess the cellular effect of silver with gallium together. That is, if the cells were affected by the presence of silver, since the positive outcome of using Ga on cells was already demonstrated [6]. For this reason, it was not thought of using the “mono-metallic samples”. 

[6] Development of novel dual-action coatings with osteoinductive and antibacterial properties for 3D-printed titanium implants. (https://doi.org/10.1016/j.surfcoat.2020.126381).

9) Although the article is titled "Synergistic effect..." no clear evidence of existing synergism was found. If the coating combines the positive properties of the components (Conclusions), then this is not synergism. Synergism necessarily includes the emergence of new properties or a fundamental change in existing ones. This is not observed here, or is not clearly presented.

The reasons for the expected synergies should also be briefly discussed.

As we explained before, the synergistic effect consist of Ga allowing to reduce the concentration of silver used, besides of providing its beneficial effect at the cellular level on the biointegration of the implant.

10) Technical Note: Calcium acetate notation as "C4H6CaO4" in Fig. 1 is confusing. Please enter the chemical formula, such as for calcium chloride.

We corrected it as Ca(CH3COO)2

Reviewer 2 Report

In this manuscript, the authors reported the synergistic effect of gallium and silver on osseointegration and antibacterial properties of calcium titanate coatings on porous titanium structures. It is a very interesting work. The experiments are good-designed and the obtained results are very nice. All the conclusions are supported by the presented data. Based on these points, this manuscript is recommended for publication at JFB.

Author Response

Reviewer 2:

In this manuscript, the authors reported the synergistic effect of gallium and silver on osseointegration and antibacterial properties of calcium titanate coatings on porous titanium structures. It is a very interesting work. The experiments are good-designed and the obtained results are very nice. All the conclusions are supported by the presented data. Based on these points, this manuscript is recommended for publication at JFB.

The authors want to acknowledge the reviewer for taking the time for the review of our manuscript and we appreciate its comments.

Round 2

Reviewer 1 Report

The authors have provided some responses to my comments; however, I do not see any corresponding changes in the text. This is true even if the Authors' Response indicates that changes have been made.

The simplest examples:

a) "2.5. Ion release: by means ICP analyses, the containing of Ga was analyzed." In the version of the manuscript I received, there is no mention of Ga in the relevant section (now it has become part 4.5).

b) "We corrected it as Ca(CH3COO)2". In the new Figure 1, the formula remains "C4H6CaO4".

I emphasize that this applies to absolutely all comments.

Moreover, in the current version, the added text (green) repeats word-by-word the crossed-out text (red). It looks like the authors create the illusion of fixes that they don't actually make. This is completely unacceptable.

Allowing the possibility of a technical error, at this stage I do not use the "Reject" function.

Thus, my current review repeats the previous one. Moreover, I believe that it is necessary to include brief comments (with links) on each of the previous comments in the text of the article. For all 10 previous comment, I expect authors to explicitly indicate the position of additional sentences in the text.

Since it became clear from the authors' answers that not all comments were correctly understood, I add explanations for them:

5) "Disk" in the remark about the lack of information about the initial morphology and morphology of the coatings, as well as about the deposition conditions, means "non-porous sample", i.e. 2D-samples.

9) The authors pointed out that Ga performs a separate function, which is not affected by silver. Moreover, the lack of a comparison of the biological properties of "bimetallic" coatings with silver analogs in one experiment does not allow us to show that the addition of Ga makes it possible to reduce the effective concentration of Ag. Moreover, it is quite likely that it is not Ga that acts, but simply a more developed surface due to the formation of a coating. Overall, there is insufficient evidence for a synergistic effect.

Author Response

The authors want to apology because not the right version of the manuscript was submitted.

Now, the final version of the article is available for the reviewer with the format of the journal.

Reviewer 1.1:

The manuscript under review is aimed at improving the biological characteristics of metal implants. This study builds on previous work of these authors (ref. 22) on the development of modified calcium titanate coatings on 3D porous titanium structures. The main idea is to include gallium and silver ions instead of silver ions only. 

Such coatings were prepared from solutions with different concentrations of gallium and silver according to a previously developed method. The samples were characterized by SEM, EDS, Raman-spectroscopy and powder XRD methods. Some in vitro biological characteristics are also presented.

In general, the manuscript contains new scientific results that meet the interests of the readers of IJMS. However, for publication it is necessary that the presentation and discussion of the results be more convincing.

The authors want to acknowledge the reviewer for taking the time for the review of our manuscript. We proceed to discuss each point:

1) In the abstract, the authors state that "The Ti surface doping was confirmed by the formation of Ga-containing Ca titanates and nanoparticles of metallic Ag within the titanate coating". The evidence presented is insufficient.  In particular, an EDS map of the distribution of metals in the coating is needed to assess the content uniformity.

We already did an EDS mapping at the time, but no particular agglomeration of any of the metal ions was observed. Thus, we understand that the sensitivity of EDS mapping is not enough to study the distribution of these ions. In a previous work [1], the presence of Ag nanoparticles was confirmed by TEM images of the cross-section of a FIB lamella of the Ti sample treated with only AgNO3. The sensitivity of EDS mapping was not enough.

[1] Bioactivity and antibacterial properties of calcium- and silver-doped coatings on 3D printed titanium scaffolds;https://doi.org/10.1016/j.surfcoat.2021.127476).

Second, the authors' statement about the formation of metallic silver nanoparticles is incomprehensible. It is not clear which method showed the presence of individual Ag nanoparticles. Their size distribution is unclear. Moreover, although the coatings contain metallic silver according to XRD data, the oxidized silver is found according to Raman spectroscopy.

Ag nanoparticles were observed and analyzed in a previous work of ours [1]. The XRD results show the same metallic silver presence observed before. We clarified this in the third paragraph in section 2.2.1. Surface characterization (Lines from 185 to 187)

Basically, according to Ellingham diagrams, the silver oxide present before heat treatment (Raman studies) decomposes at temperatures above 200 ◦C. Therefore, after the thermal treatment of the process when 600 ◦C was applied, the metallic silver appeared as a product of the decomposition of Ag2O. Consequently, no Ag oxides were observed in the XRD spectrum after thermal treatment.

2) The representation of the composition of the coatings according to the EDS data looks incorrect (Figs. 4b, 3b).  First, it is not clear how the analyzer used can produce such low atomic content errors (e.g, 34.7±0.4 oxygen). This requires a number of special calibrations. No relevant information provided. Without calibration, the errors given by the software are only indicative. In fact, such errors are then not less than 5-10% of the determined value.

Secondly, values of type "0.5±4" (calcium content in Fig. 4b) have no physical meaning. Here, the "determination error" exceeds a certain value. It's just incorrect data. 

We agree with the reviewer about the values of the standard deviations, and we corrected the errors (Figure 2b line 138 and Figure 3b line 200). The objective of EDS here is only to identify the incorporation of Ga and Ag ions in the titanate surface and to have an idea of the quantity. We are aware of the error of the technic and that’s why the results are semiquantitative. However, we could confirm that, for example, when an element was doubled in the initial concentration applied, the detection increases with the EDS.

Moreover, what is the reason for such a low content of calcium, which does not correspond to the stoichiometry of titanate? I mean, why is the calcium content much less than the content of the doping component, for example, silver? Initially, the amount of calcium introduced was at least an order of magnitude greater than that of silver. Explain the reason for this discrepancy also in the text of the manuscript.

It is not a stoichiometric structure, since the ions are incorporated into the interstices of the titanates [2]. It is a deficient structure of Ca, Ag and Ga, due to the last process of heat water, which improves their release.

[2] Takashi Kizuki et al. Preparation of bioactive Ti metal surface enriched with calcium ions by chemical treatment doi:10.1016/j.actbio.2010.01.007

3) The absolute absence of carbon in the composition of the coating raises doubts. In fact, such impurities may remain with the chosen method of preparation. It is required to add the EDS spectra in the corresponding area in the Supplementary Materials.

We eliminate the carbon, to have the absolute values of the ions to study. The values we obtained for C were between 1 and 0.5 % but considering the errors it is not significant. EDS helps us to have an approximation.

4) The effect of doping components on the morphology of coatings is not discussed. However, the data in Figure 5a suggest that there is some effect. Give particle sizes on the surface and a description of the morphology, note the thickness of the coatings.

It was discussed but in section 2.2.1.: “Electron microscopy evaluation of the Ti surfaces treated with the different concentrations of Ag and Ga nitrates showed typical porous acicular feather-like structures (Figure 4a).” And we incorporated the following sentence: “Thus, the incorporation of metal ions does not affect the typical topography of the treatment “(Line 161).

We have to clarify that the data in Figure 5a (now is Figure 4a) correspond to the images of the (simulated body fluid) SBF essay, showing a Hydroxyhypatite coating, which is commented on in section 2.3. Ion release and cytotoxicity (Line 234)

5) The data in Figure 5b looks incorrect. The axes need to be fixed, now they look artificially drawn and do not match the graphs.

The authors considered the increase of the axes’ letter size to adapt better the edition of the paper and facilitate its reading. Logically, if necessary, it can be changed.

It is necessary to explain why "whiskers of error" are present both on the vertical axis (metal content) and on the horizontal axis (time). Should it be understood that in the experiments it was not possible to clearly fix the time?

In principle, time has to be fixed. However, in some cases due to experimental errors, there were time variations that we wanted to reflect in the graph.

It is necessary to rid the text of the manuscript of contradictions with the data of the Figure. In fact, judging by the Figure, silver release is not much different for porous and non-porous samples (see scales). It is clear from the text that it is different (limes 345-346). 

We introduce this observation in the text. “In general, the Ag and Ga release from porous 3D Ti structures was favored due to the greater specific surface of the samples compared to the discs, with the exception of the CaGa5Ag1 sample, which is approximately the same release from both 2D and 3D samples.” (Lines 236-239)

On the other hand, it follows from the text that saturation (stabilization) is not reached only for calcium release (line 350). However, the purple silver release curve also did not reach saturation. It's not explained.

We introduce the observation in the test: “Regarding the silver release, the porous sample treated with the maximum silver nitrate concentration (CaGa5Ag5) showed greater ion release and its curve did not stabilize after 48 hours. This sample was the only one showing less than 70 % cell viability in the toxicity test (Figure 4c). As it was considered cytotoxic, this sample was discarded in the cell study”. (Line 246-50)

Comparison of ion release kinetics with literature (lines 347-349) is unclear. Clarification of the experimental conditions is required.

The cited articles are works of ours and were tested under the same conditions according to the ISO-10993-12 standard. Thus, they are comparable.

We clarify this in the test: “The Ti surfaces produced with the higher concentration of Ga and Ag ions exhibited greater release of the respective ions agreeing with the literature where similar studies were carried out in the same conditions (ISO-10993-12 standard),applying Ga [5] and Ag [22] separately.“ (Line 239-242)

Finally, explain how the gallium content of the solution was measured and how the samples were prepared on the disks. What was the initial morphology of the disks and what was the morphology of the coatings on them. Provide SEM data. Otherwise, the comparison is incorrect.

This is all in M&M:

  • 5. Ion release: by means ICP analyses, the containing of Ga was analyzed.
  • 2. Surface thermochemical treatment: here, you will find how the samples with Ga were prepared on disks.
  • Figure 3 and 4: the morphology of the surfaces treated with ga is shown.

6) As far as is clear from the experimental part, сell viability was examined after 24-h exposures. Thus, the explanation for the toxic effect observed for sample CaGa5Ag5 is incorrect (lines 354-355). In fact, after 24 hours, the silver concentration there is no higher than in other samples (Fig. 5b). Therefore, offer a more correct explanation.

In the viability test, the samples with the thermochemical treatment were incubated for 72h in Dulbecco's modified Eagle's medium (DMEM). Then, cells were cultured in this medium. This means that the ion released to which cells were exposed was the equivalent of 72h. Probably, the continuous release of silver ions can produce toxicity.

In addition, specify the limits of the safe concentration of each metal in biological solutions (in ppm, according to the literature data and compare with the content being achieved.

There is a consensus on the MIC Ag ions values for different bacterial strains. In our case, we exceed the MIB values reported in the literature. The release curves show that the silver ion values at 24h oscillate between 0.8-1 ppm (mg/L). In all cases, we exceed the MIC values reported in the literature; Staphylococcus aureus MRSA [3] (0.84 mg/L), Pseudomonas aeruginosa [4] (0.84-0.85 mg/L),Escherichia Coli [4] (1 mg/L) and Staphylococcus epidermidis [4] (0.5 mg/ L).

[3] Libor Kvitek et al. Antibacterial activity and toxicity of silver- nanosilver versus ionic silver DOI:10.1088/1742-6596/304/1/012029

[4] Wen-Ru Li et al. A comparative analysis of antibacterial activity, dynamics, and effects of silver ions and silver nanoparticles against four bacterial strains. DOI: 10.1016/j.ibiod.2017.07.015

7) According to Figure 6a, the sample CaGa5 as well as CaAg1 has a more pronounced antibacterial effect towards to P. aeruginosa than all the bimetallic-contained samples. Thus, a synergistic effect is manifested in a decrease in bactericidal action. Now this is not consistent with what is written in the text. Please bring the information in line with the data and explain the possible reasons for this effect. 

The growth inhibition halo assay is approximate and is related to the diffusion of the metallic ion through the agar. Thus, the results from this essay give us some information about the metallic ion antibacterial capability. The small differences in halo diameters are not significant. We clarify this in the test. (Lines 273-275)

In Figure 6b, there are no data for "monometallic-doped " samples (CaGa5 and CaAg1). In general, from the results of the figure it seems that the "bimetallic-doped" samples (CaGaXAgY) are noticeably less efficient than the monometallic ones

We agree with the reviewer, we should have used the controls of both ions separately. However, with these tests, we can observe the synergy between the two incorporated elements. Ga shows a great antibacterial effect with respect to the resistant strain P. aerugiosa(Figure 5b): with the same amount of silver and more gallium (compare sample CaGa5Ag2 blue with CaGa10Ag2 light green in the graph), the inhibitory effect increases. Furthermore, with less silver amount and the same amount of gallium, the antibacterial effect is considerably improved (compare to samples CaGa5Ag1 dark green and CaGa5Ag2 blue). The latter gives us proof that when gallium is present, we can reduce the amount of Ag (important due to its possible toxicity).

Probably when carrying out the study of bacterial adhesion at such a short time of 2h, a clear synergistic effect is not appreciated. The antibacterial effect of gallium ions is less than in the case of silver. For example, the MIC values reported for gallium ions are higher than for silver; for example E. Coli (256 mg/L), Pseudomonas aeruginosa, P. aeruginosa and Staphylococcus aureus aprox. 512 mg/L (5).

In terms of a medical device, the possible toxicity of silver is a concern at the regulatory level. As the inhibitory power of Gallium is less than silver, the aim of using the synergy of both metallic ions is to be able to reduce the concentration of silver needed. Moreover, Ga has other advantages as to stimulate bone formation by acting on osteoblast cells and is already on the US FDA approved list. (Lines 296-300)

[5] Antimicrobial effect of gallium nitrate against bacteria encountered in burn wound infections. DOI: 10.1039/C7RA10265H

8) Similarly, section 3.5. Cell Response does not provide comparative data on "monometallic-doped" coatings (CaGa5 and CaAg1). This does not allow us to support the idea and final conclusions of the article about the preference for "bimetallic-doped" coatings.

Yes, we could indeed have tested the two metals separately. However, this assay was carried out to assess the cellular effect of silver with gallium together. That is, if the cells were affected by the presence of silver, since the positive outcome of using Ga on cells was already demonstrated [6]. For this reason, it was not thought of using the “mono-metallic samples”. 

[6] Development of novel dual-action coatings with osteoinductive and antibacterial properties for 3D-printed titanium implants. (https://doi.org/10.1016/j.surfcoat.2020.126381).

9) Although the article is titled "Synergistic effect..." no clear evidence of existing synergism was found. If the coating combines the positive properties of the components (Conclusions), then this is not synergism. Synergism necessarily includes the emergence of new properties or a fundamental change in existing ones. This is not observed here, or is not clearly presented.

The reasons for the expected synergies should also be briefly discussed.

As we explained before, the synergistic effect consists of Ga allowing to reduce the concentration of silver used, besides of providing its beneficial effect at the cellular level on the biointegration of the implant.

10) Technical Note: Calcium acetate notation as "C4H6CaO4" in Fig. 1 is confusing. Please enter the chemical formula, such as for calcium chloride.

We corrected it as Ca(CH3COO)2 (line 99-103)

Reviewer 1.2:

The authors have provided some responses to my comments; however, I do not see any corresponding changes in the text. This is true even if the Authors' Response indicates that changes have been made. 

The simplest examples:

  1. a) "2.5. Ion release: by means ICP analyses, the containing of Ga was analyzed." In the version of the manuscript I received, there is no mention of Ga in the relevant section (now it has become part 4.5).
  2. b) "We corrected it as Ca(CH3COO)2". In the new Figure 1, the formula remains "C4H6CaO4".

I emphasize that this applies to absolutely all comments.

Moreover, in the current version, the added text (green) repeats word-by-word the crossed-out text (red). It looks like the authors create the illusion of fixes that they don't actually make. This is completely unacceptable.

Allowing the possibility of a technical error, at this stage I do not use the "Reject" function.

Thus, my current review repeats the previous one. Moreover, I believe that it is necessary to include brief comments (with links) on each of the previous comments in the text of the article. For all 10 previous comments, I expect authors to explicitly indicate the position of additional sentences in the text.

Since it became clear from the authors' answers that not all comments were correctly understood, I add explanations for them:

5) "Disk" in the remark about the lack of information about the initial morphology and morphology of the coatings, as well as about the deposition conditions, means "non-porous sample", i.e. 2D-samples.

The morphology obtained in solid Ti as in porous Ti is the same as already verified in previously published articles [1,6]. In addition, some of the techniques used can only be performed with a solid, non-porous material (discs), such as low-angle X-ray spectroscopy.

9) The authors pointed out that Ga performs a separate function, which is not affected by silver. Moreover, the lack of a comparison of the biological properties of "bimetallic" coatings with silver analogs in one experiment does not allow us to show that the addition of Ga makes it possible to reduce the effective concentration of Ag. Moreover, it is quite likely that it is not Ga that acts, but simply a more developed surface due to the formation of a coating. Overall, there is insufficient evidence for a synergistic effect.

Our understanding of synergy in our case is that silver provides antibacterial properties that are enhanced by the presence of gallium (for the p. aeruginosa strain). In addition, the presence of gallium can allow reducing the silver content (which is positive due to the possible toxicity of silver in high concentrations). In addition, the presence of gallium improves the osseointegration and mineralization of the material.

Surely, more tests need to be carried out, but in these preliminary tests, the trend that we clearly see is these.

Round 3

Reviewer 1 Report

The actual version of the article reflects the necessary changes; however, a number of key points still need to be adjusted.

1) The "synergism" is still not proven. Ga performs his own function, on which the introduction of silver does not affect. In fact, evidence has not been provided that the introduction of gallium allows reducing silver concentration sufficient for the necessary antibacterial effect. Figure 5A does not have a difference between the properties of the samples of the CaAg1 and CaGaXAgY. According to Figure 5b, there is no significant difference between samples CaGa5Ag1 and CaGa5Ag2, as well as there is no comparison with the corresponding CaGa5 and CaAg1 or CaAg2 samples. This does not allow us to draw a conclusion indicated by the authors on line 519 (yellow).

Thus, the authors should either carry out all the necessary studies, or remove the unproven formulations from everywhere, starting with the Title. In other words, "compositional material" - yes, "synergistic effect" – no, since it is prematurely.

2) Data on the release of metals still raise questions. In particular, in the case of calcium, the concentration axes look artificially drawn. In fact, it is expected to see the marks of a monotonous increase in the concentration from the type "0 => 0.1 => 0.2 => 0.3 => 0.4", while the incomprehensible " 0 => 0.1 => 0.1 => 0.2 => 0.3 => 0.3 "is presented. The marks duplicate each other, and the concentration is growing on the graph! This is impossible. It makes me doubt all the data from this section.

In addition, ICP does not show the shape of the metal being determined, so it is difficult to say that we are talking about gallium, silver and calcium ions. At a minimum, these are aquated ions, but rather, chemically more complex forms.

3) Regarding the morphology of the samples, Figure 5 clearly shows that in the case of CaGa5Ag5 and CaGa5Ag1 samples, the formation of bigger agglomerates on the surface is observed, which differ markedly from the rest. Give a full description and comparison with the corresponding "monometallic" samples CaGa5 and CaAg1 and CaAg5.

In addition, the coincidence of the morphology of the samples on 2D and 3D titanium should be noted. SEM images of the surface would be preferred for visual comparison. They can be given in Supplementary Materials, but the reader should not be forced to look for evidence himself from the previous works of the authors. In addition, it seems unlikely that the thickness of the coatings on such objects would match.

Author Response

Reviewer 1.3.

The actual version of the article reflects the necessary changes; however, a number of key points still need to be adjusted.

1) The "synergism" is still not proven. Ga performs his own function, on which the introduction of silver does not affect. In fact, evidence has not been provided that the introduction of gallium allows reducing silver concentration sufficient for the necessary antibacterial effect. Figure 5A does not have a difference between the properties of the samples of the CaAg1 and CaGaXAgY. According to Figure 5b, there is no significant difference between samples CaGa5Ag1 and CaGa5Ag2, as well as there is no comparison with the corresponding CaGa5 and CaAg1 or CaAg2 samples. This does not allow us to draw a conclusion indicated by the authors on line 519 (yellow).

Thus, the authors should either carry out all the necessary studies, or remove the unproven formulations from everywhere, starting with the Title. In other words, "compositional material" - yes, "synergistic effect" – no, since it is prematurely.

Following the reviewer's advice, the authors decided to make the changes in regard to the synergistic effect that we referred, and we changed the title and the content, eliminating the concept of synergy. We have also removed the phrase from line 519.

The synergistic words have been removed so as not to be misleading.2) Data on the release of metals still raise questions. In particular, in the case of calcium, the concentration axes look artificially drawn. In fact, it is expected to see the marks of a monotonous increase in the concentration from the type "0 => 0.1 => 0.2 => 0.3 => 0.4", while the incomprehensible " 0 => 0.1 => 0.1 => 0.2 => 0.3 => 0.3 "is presented. The marks duplicate each other, and the concentration is growing on the graph! This is impossible. It makes me doubt all the data from this section.

This was a typing error. We corrected the axis number in the graphic.

In addition, ICP does not show the shape of the metal being determined, so it is difficult to say that we are talking about gallium, silver and calcium ions. At a minimum, these are aquated ions, but rather, chemically more complex forms.

The Ca2+, Ga3+ and Ag+ ions have been incorporated into the graph for greater clarity.

3) Regarding the morphology of the samples, Figure 5 clearly shows that in the case of CaGa5Ag5 and CaGa5Ag1 samples, the formation of bigger agglomerates on the surface is observed, which differ markedly from the rest.

In the SEM images in figure 5a, the authors do not observe any differences. All samples show the typical porous acicular feather-like structures.

It is true that in previous works, where the surface was doped only with silver, we have observed the increase in the thickness of the feathers in a subtle way. However, when the two elements, Ga and Ag, are incorporated, this effect is not observed.

We incorporated this in the text “In previous papers, when the surfaces were only doped with silver, a slight increase in the thickness of the feathers of the structure was observed [24]. However, in the case of the simultaneous incorporation of both Gallium and Silver, this effect is not observed.”

Give a full description and comparison with the corresponding "monometallic" samples CaGa5 and CaAg1 and CaAg5.

The SEM image of the sample CaGa5 is shown in figure 2a.

The CaAg1 sample is published the article [1] Bioactivity and antibacterial properties of calcium- and silver-doped coatings on 3D printed titanium scaffolds; https://doi.org/10.1016/j.surfcoat.2021.127476).

The SEM image of the CaAg5 sample is not published (x50,000) and it is observed that it is the same structure as for the CaAg1 case:

What really interests us, more than the topography, is the bioactivity that is generated in contact with SBF. In this work, this bioactivity is confirmed for samples with the two ions together. With the individual ions, the bioactivity has been confirmed in the two previous studies already published.

In addition, the coincidence of the morphology of the samples on 2D and 3D titanium should be noted. SEM images of the surface would be preferred for visual comparison. They can be given in Supplementary Materials, but the reader should not be forced to look for evidence himself from the previous works of the authors. In addition, it seems unlikely that the thickness of the coatings on such objects would match.

Regarding the morphology of the titanate, this is practically the same in the case of 2D and 3D (attached SEM images). As they are so similar, we do not believe it is necessary to incorporate them as supplementary material. However, we do not have any problem if the editor considers it important.

2D 3D

Round 4

Reviewer 1 Report

In its present form, the manuscript does not contain obvious contradictions / shortcomings, therefore, it can be accepted for publication. I hope that the authors will develop the issue of "synergism" in subsequent studies, because this topic is very interesting.

The corrections required prior to publication:

(1) in Figure 5b and throughout the text, it is necessary to replace "Ca ion" with "Ca". The same should be done for other metals. In fact, the method used (ICP-AES) only allowed the authors to determine the metal concentrations. However, it did not allow to determine the chemical form in which these metals are existed. Taking into account the composition of the solution, these definitely cannot be "free" ions (Ca2+, Ga3+, Ag+), but these are chloride or aqua complexes, associates with glucose, and so on. The authors did not study this issue, so more general terminology should be used.

(2) A description of the difference in morphology obtained (after exposure to SBF) should be added. In response to previous comments, the authors referred to other figures (2 and 3). The calcium phosphate crystals in Figure 4a, deposited on different samples, clearly differ in size. Or is this the "standard" size variation for the given experimental conditions?

Author Response

Reviewer 1.4.

In its present form, the manuscript does not contain obvious contradictions / shortcomings, therefore, it can be accepted for publication. I hope that the authors will develop the issue of "synergism" in subsequent studies, because this topic is very interesting.

The corrections required prior to publication:

(1) in Figure 5b and throughout the text, it is necessary to replace "Ca ion" with "Ca". The same should be done for other metals. In fact, the method used (ICP-AES) only allowed the authors to determine the metal concentrations. However, it did not allow to determine the chemical form in which these metals are existed. Taking into account the composition of the solution, these definitely cannot be "free" ions (Ca2+, Ga3+, Ag+), but these are chloride or aqua complexes, associates with glucose, and so on. The authors did not study this issue, so more general terminology should be used. 

We changed the graphic with “Ca concentration (ppm)”, “Ag concentration (ppm)” and “Ga concentration (ppm)” and eliminated the word ion in the test.

(2) A description of the difference in morphology obtained (after exposure to SBF) should be added. In response to previous comments, the authors referred to other figures (2 and 3). The calcium phosphate crystals in Figure 4a, deposited on different samples, clearly differ in size. Or is this the "standard" size variation for the given experimental conditions?

We added a paragraph discussing the morphology of the calcium phosphate precipitates (lines 232 to 238).
